# Automated Assessment of Digital Images of Uterine Cervix Captured Using Transvaginal Device—A Pilot Study

**DOI:** 10.3390/diagnostics13193085

**Published:** 2023-09-28

**Authors:** Saritha Shamsunder, Archana Mishra, Anita Kumar, Sachin Kolte

**Affiliations:** 1Gynecology Department, Safdarjung Hospital, New Delhi 110029, India; drarchanamishra@rediffmail.com (A.M.); anitakumar8@yahoo.com (A.K.); 2Department of Pathology, VMMC and Safdarjung Hospital, New Delhi 110029, India; drsachinkolte@gmail.com

**Keywords:** point-of-care test, cervical cancer screening, digital VIA-VILI, artificial intelligence, auto-assessment

## Abstract

In low-resource settings, a point-of-care test for cervical cancer screening that can give an immediate result to guide management is urgently needed. A transvaginal digital device, “Smart Scope^®^” (SS), with an artificial intelligence-enabled auto-image-assessment (SS-AI) feature, was developed. In a single-arm observational study, eligible consenting women underwent a Smart Scope^®^-aided VIA-VILI test. Images of the cervix were captured using SS and categorized by SS-AI in four groups (green, amber, high-risk amber (HRA), red) based on risk assessment. Green and amber were classified as SS-AI negative while HRA and red were classified as SS-AI positive. The SS-AI-positive women were advised colposcopy and guided biopsy. The cervix images of SS-AI-negative cases were evaluated by an expert colposcopist (SS-M); those suspected of being positive were also recommended colposcopy and guided biopsy. Histopathology was considered a gold standard. Data on 877 SS-AI, 485 colposcopy, and 213 histopathology were available for analysis. The SS-AI showed high sensitivity (90.3%), specificity (75.3%), accuracy (84.04%), and correlation coefficient (0.670, *p* = 0.0) in comparison with histology at the CINI+ cutoff. In conclusion, the AI-enabled Smart Scope^®^ test is a good alternative to the existing screening tests as it gives a real-time accurate assessment of cervical health and an opportunity for immediate triaging with visual evidence.

## 1. Introduction

Cervical cancer is the fourth most common cancer in women. According to statistics from 2020, the global incidence rate of cervical cancer is 15.6/10^5^ (604,127) with mortality at 8.8/10^5^ (341,831) [1]. In low- and middle-income countries, cervical cancer is the second most common cancer among women, with 236,828 new cases in 2020. It claimed 146,198 deaths with a five-year prevalence rate of 35.00/10^5^ [2]. In India, the incidence is at 18.7/10^5^ (123,907), while mortality is at 11.7/10^5^ (77,348) [3]. Access to screening and health services is a major barrier for success in cervical cancer screening programs in low- and middle-income group countries. In India, fewer than 1 in 10 women have been screened for cervical cancer in the last 5 years [4]. The WHO has called for the elimination of cervical cancer by 2023 through the vaccination, screening, and treatment of women. As per the recent WHO guidelines, a screen-and-treat approach using HPV DNA detection is recommended [5]. Considering its high infrastructure cost and the requirement of a supplementary test for the visualization of the lesion for treatment, VIA seems to be an economically viable alternative in resource-constrained settings for a single-visit result approach [6]. However, VIA is highly subjective with high false positive and false negative rates. Therefore, colposcopy is commonly performed for triaging while colposcopy-guided biopsy remains a critical diagnostic step. However, the colposcope is a bulky, expensive, electricity-dependent, and externally mounted high-maintenance device not suitable for community settings. Therefore, triaging is dependent on the patient traveling to tertiary facilities as of today. In most low-resource settings, this leads to a loss in follow-up and treatment. It is vital to reduce the dependence on colposcopy-based triaging by using a point-of-care test and implement a screen-and-treat strategy to reduce ‘loss to follow-up’ cases. The documentation of cervical images is also essential for quality assurance and peer review in case of any doubt, before further improvising the incorporated AI system.

In view of these requirements, a transvaginal digital device, “Smart Scope^®^” (SS), was manufactured by Periwinkle Technologies Pvt. Ltd., India. The Smart Scope^®^-aided VIA test (SS Test) was compared with Pap smear and naked eye VIA screening tests by a team of gynecologists and pathologists at a tertiary care center in Pune, India [7]. The sensitivity and NPV of the SS test was demonstrated to be 100% each, which was significantly higher than the Pap and naked eye VIA, while the specificity of the SS test was 36.8% [7]. 

To enable such triaging in community settings, an auto-image-assessment feature enabled by artificial intelligence (AI) was introduced in the software platform of Smart Scope^®^. The AI-based auto-assessment (SS-AI) categorizes the cervix into four groups based on the risk assessment. 

The present study was aimed at validating the AI-enabled automated assessment (SS-AI) of digital images captured by the transvaginal device, Smart Scope^®^, as a primary method for cervical health screening.

## 2. Materials and Methods

### 2.1. Device (Smart Scope^®^)

Smart Scope^®^ (model CX2.0, Periwinkle Technologies Pvt. Ltd., Pune, India) is a portable, opto-digital, transvaginal device (Figure 1) attached to a tablet with the pre-installed Net4Medix^®^ Version 53 software (Periwinkle Technologies Pvt. Ltd., Pune, India) for the storage of patient demographic data and cervix images. 

The AI model (Smart Scope^®^ CX Version 3) implemented in this study has been trained and internally validated by Periwinkle Technologies using 150,000+ images captured by Smart Scope^®^ all over India. The machine learning model in the software classifies the images into one of four classes. Based on the risk level detected by the SS-AI, the images were grouped into four categories as follows: green (normal cervix, Nabothian cyst); amber (polyp, infection, inflammation, cervicitis, squamous metaplasia); high-risk amber (HRA, probable low-grade CIN lesion); and red (probable high-grade CIN lesion and carcinoma).

### 2.2. Methods

A prospective observational, single-arm, non-randomized study was carried out in the Obstetrics and Gynecology OPD of Safdarjung hospital, a government tertiary care center, after the Institutional Ethics Committee approval (Sr. no. IEC/VMMC/SJH/Project/2021-03/CC-129, dated 17 May 2021) and CTRI registration (CTRI/2021/07/035164). The gynecologist and the expert colposcopist were blind to each other’s assessment as well as that of the AI. Healthy, non-pregnant eligible women who were sexually active, coming for any gynecological problem with an intact uterus and no history of cervical pre-cancer or cancer were counselled. Women who had undergone previous treatment for CIN, had a frank cervical growth, or not willing for follow-up were excluded from the study. Active cervicitis and vaginitis was treated before the test. Consenting women were enrolled into the study. 

The demographic information of all enrolled women was recorded in the software (Net4Medix^®^ Version 53) provided in tablet. All women underwent cervical screening using the transvaginal digital device (Smart Scope^®^)-aided VIA test performed by a nurse under the supervision of a gynecologist (SS-VIA) [8,9]. During the test, the cervix was cleaned using normal saline and images of cervix were captured. Following this, 5% acetic acid and Lugol’s iodine were applied sequentially using a cotton swab and images were captured 1 min after acetic acid application and immediately after Lugol’s iodine. 

The captured images were assessed by the cloud-based machine learning model after establishing Internet connectivity. The AI classed the images into the four categories and the corresponding color code was immediately displayed by clicking the “Assess” button on the tablet. High risk amber and red were categorized as screen positive on AI assessment. All the SS-AI screen positive women (red and HRA) were counselled and called for colposcopy. The cervical images of SS-AI that were negative for women (green and amber) were evaluated by an expert colposcopist (SS-M) on the Net4Medix^®^ portal. Those suspected of being positive (with a minor lesion or worse) were called for colposcopy.

Colposcopy was carried out by trained colposcopists with a digital video colposcope in the colposcopy clinic situated in the outpatient area. In lithotomy position, the woman’s cervix was visualized after sequentially applying normal saline, 5% acetic acid, and Lugol’s iodine. Colposcopy was reported as normal or abnormal. Abnormal (Swede score 1 and above) colposcopy findings were graded as minor, major, or carcinomas. Biopsy was taken as the gold standard. Where biopsy was not available, the expert assessment of Smart Scope^®^ images or colposcopy assessment was taken as the reference standard.

### 2.3. Statistical Analysis

For statistical analysis, women categorized as red and HRA on the AI assessment, indicating a suspected low-grade lesion or worse according to the experts’ assessment (SS-M), were considered as positive. Colposcopies of women with a Swede score of 1 and above were considered positive. Histopathology was considered the gold standard with CIN 1+ as the cut off. Where the histopathology outcome was not available, the colposcopy assessment was considered to be the reference standard. 

Statistical analysis was carried out with the help of SPSS (version 20) for Windows package (SPSS Science, Chicago, IL, USA). The description of the data was performed in the form of an arithmetic mean ± SD for quantitative (continuous) data and in the form of frequencies (%) for qualitative (categorical) data. *p*-values of <0.05 were considered significant.

The chi-square test which indicates the association between the two tests in question was applied for assessing the performance of SS-AI against histopathology and colposcopy. The accuracy was described by the following parameters: sensitivity, specificity, positive predictive value (PPV), negative predictive value (NPV), and area under the curve (AUC) at a confidence interval (CI) of 95. 

## 3. Results

A total of 927 women were enrolled in this study. Enrolled women for whom AI assessment could not be performed or resulted in poor-quality images were not considered for further analysis. Following this cleanup, data on SS-AI were available for 877 women. Colposcopy and biopsy were performed on 485 and 213 women, respectively. 

The demographic data distribution is given in Table 1. Most (50%) women who participated in this study were in the reproductive age group of 25–35 years. The mean age of the participants was 37 years. Only 17% of the women had completed graduation or postgraduation; primary school was the highest level of education obtained by 10% of the women; middle school was the highest level of education obtained by 20% of the women; secondary school was the highest level of education obtained by 20% of the women; and 33% were uneducated. More than 97% women were housewives, and the majority of these women were of low socio-economic status. Sexual intercourse usually begins at the time of marriage in India; about 27% women were married and became sexually active before the age of 18 years. The majority (40.25%) of women had two children; 27.37% of women had three children with a mean parity of two.

On the SS-AI assessment of 877 women, 366 (41.73%) were found to be positive while 511 (58.27%) were negative. (Figure 2). 

Out of the SS-AI positives, 93 women did not revisit the hospital for further colposcopy assessment (loss to follow-up). The remaining 273 had colposcopy in the colposcopy clinic; biopsy was advised for 136 women with a Swede score ≥1. Histopathology was available for 134 women as 2 refused to have a biopsy. By means of histopathology, 81 were determined to have CIN I, 20 were determined to have CIN II, 9 were determined to have CIN III, while 2 were determined to have carcinoma. Thus, upon an SS-AI positive outcome followed by colposcopy triaging, 83.58% (112/134) of the women were found to have precancerous or cancerous lesions (Figure 2). 

There were 511 SS-AI negative women. The cervical images of these women were assessed by the expert colposcopist. Upon assessing the digital images of 511 SS-AI-negative women (Figure 2), the expert categorized (SS-M) 220 (43%) women as positive, among which 212 women attended the colposcopy clinic. Colposcopy report was normal in 131; biopsy was taken in 81 women who had an abnormal colposcopy. By means of histopathology, 66 women were negative for precancer or cancer, 8 women had CIN I lesions, 2 had CIN II, while 2 women had carcinomas. Thus, upon an SS-AI negative outcome followed by expert’s digital image assessment and colposcopy triaging, 14.8% (12/81) women were found to have precancerous or cancerous lesions. 

Colposcopy was carried out for 485 women [273 (SS-AI positive) + 212 (SS-AI negative but SS-M positives)]. In terms of colposcopies, 267 women had normal outcomes, while 218 women had abnormal findings (Figure 3). Minor grade lesions (Swede score 1–4) were found in 124 women, 92 had major grade lesions (Swede score ≥ 5), and 2 were found to have carcinoma. The histopathology outcome was available for 211 women. Upon histopathology, 87 were negative for precancer or cancer, 89 had CIN I, 22 had CIN II, 9 had CIN III lesions, while 4 had carcinoma (Figure 3). Thus, upon colposcopy triaging followed by histopathology, 58.21% (124/213) women were found to have precancerous or cancerous conditions. 

Performance of SS-AI against histopathology outcome was based on results on a sample size of 213 biopsy results (Table 2). With SS-AI at HRA+ positivity and histopathology at CIN I+ positivity, the chi-square value for SS-AI was 95.6% (*p* = 0.000), indicating that there was positive association between the SS-AI and histopathology. The SS-AI showed high sensitivity (90.3%), specificity (75.3%) as well as an excellent AUC [0.828(0.767–0.889) at CI 95, *p* = 0.000] (Figure 4). The PPV and NPV were 83.58% and 84.81%, respectively, along with good accuracy (84.04%) and a coefficient of correlation 0.67, *p* = 0.00.

Similarly, the performance of colposcopy was evaluated against the histopathological assessment of 213 women (Table 3). With the colposcopy cutoff at Swede score 1 and histopathology cutoff at CIN I, the sensitivity, specificity, PPV, and NPV of colposcopy were 100%, 2.2%, 58.77%, and 100%, respectively. The accuracy was 59.15% while the AUC was 0.51 [(0.43–0.59), *p* = 0.02, CI 95]. When the cutoff of the colposcope positivity was set at a Swede score of 5, the sensitivity, specificity, PPV, and NPV of colposcopy was 48.4%, 64%, 65.22%, and 45.97%, respectively; the histopathology detection at a CIN I+ level had an accuracy of 54.93% and an AUC of 0.56 [(0.484–0.640) *p* = 0.122, CI 95]. 

## 4. Discussion

The application of AI in the field of medicine is rapidly increasing. In gynecologic oncology, findings by various researchers on the application of AI in cervical screening have been published [10]. 

Cytology-based screening has witnessed automation in slide preparation and slide reading using computational methods [11,12]. Various conventional neutral networks (CNN) and deep learning (DL) models have been employed for the automated analysis of Pap images to improve sensitivity and reduce referrals for colposcopy [13,14,15]. Although automation and AI have improved the sensitivity of cytology, procedural delays preclude it from being a point-of-care test. 

Visual inspection with acetic acid (VIA) is an excellent point-of-care test, though the accuracy depends on the experience of the health worker. The accuracy of VIA has improved by using digital devices which allow a clear and close view of cervix. Based on the data of 62 biopsies, a stand-mounted digital device Gynocular showed 70% agreement with colposcopy [16]. At CIN II, the cutoff sensitivity of Gynocular was in the range of 71.2% (306 biopsies)–81.3% (652 biopsies) in various trials [17,18]. A transvaginal device Pocket Colposcope showed 71.2% sensitivity and 57.5% specificity, where the gold standard was histopathology (129 enrolments, 81 biopsies) [19].

The automated analysis of VIA images using AI can reduce the subjectivity and increase the accuracy of VIA. A pioneering study from Costa Rica developed AI, based on images captured using conventional analog cameras [20] which significantly improved the accuracy (cervigram with AUC 0.69, cytology with AUC 0.71, AI with AUC 0.91). Digital images of the cervix have also been used for developing AI. The AI-aided Pocket Colposcope examination showed 81.3% sensitivity and 78% specificity where the gold standard was histopathology (134 biopsies) [21].

AI is being studied as an auxiliary diagnostic tool in low- and middle-income countries (LMICs) (i) to enhance the accuracy of a visual screening test; (ii) as a triaging method; (iii) for the classification of colposcopy images, etc. [22,23,24,25,26]. Standalone algorithms which can be used along with any other android devices such as smartphones were also developed by researchers to detect precancerous lesions [27,28,29]. Enhanced visual assessment (EVA) is one such algorithm tested for triaging screen positive women [30,31,32,33,34]. An algorithm developed by Kudva and co-workers claimed an accuracy of 97.94%, a sensitivity of 99.05%, and a specificity of 97.16% in the lab test environment [35].

In the present study, at CIN I cutoff, AI demonstrated 90.3% sensitivity with high accuracy and an AUC of 0.828. This AI algorithm could accurately identify true CIN lesions, indicating that it can be effectively used as a screening device. A trained nurse at the peripheral health center can screen and identify abnormal cases in the same sitting using this AI. 

SS-AI also has potential as a triage method after HPV or Pap screening, as its accuracy for predicting CIN (84.04%) was found to be higher than that of colposcopy (59.15%). Imaging and triaging using a single platform (AI-integrated Smart Scope^®^ test) to identify true positives at peripheral health centers (PHCs) will eliminate the necessity of long-distance travel for a colposcopy examination and minimize loss to follow-up rates. Those positive on triage can be treated at the peripheral center using simple ablative methods, making the WHO goal [4] of treating 90% of CIN lesions possible.

Published studies on AI have used standard colposcopy images and/or the digital camera images of the patients who are screen positive on one of the standard screening tests viz. Pap, VIA, HPV. These AIs were not tested upon a population that was screen negative or never screened. These AI models were mainly developed with the aim of (i) triaging the pre-identified screen positive women for biopsy or (ii) proving the equivalence of the AI-enabled digital device with the standard colposcope. Thus, these AI models gave a binary outcome where CIN2 and above cases were grouped in “positive” category and the rest of the cases were grouped into a “negative” category.

The present study AI was developed solely based on the images captured using the transvaginal digital scope “Smart Scope^®^”. The AI-based Smart Scope^®^ test has been developed primarily as a screening tool and not as a triaging tool. Unlike the rest of AIs which classify the cervix in binary categories, this AI categorizes cervical health in four groups, viz. green (normal), amber (benign conditions such as cervicitis, inflammation, etc.), high-risk amber (low grade/CIN I), and red (high grade+/CIN II+). In routine screening, as per the Government of India guidelines, a woman declared as “negative” will not be asked to come for any follow-up visit for the next 3–5 years, while a woman with a low grade lesion is asked to return after one year. In addition to cancer and pre-cancer, SS-AI helps to identify cervicitis and inflammation which need timely treatment so as to improve women’s health.

The major strength of this study is the significant number of women which are not “pre-screened” by any method. The AI-based auto-assessment is highly accurate with a very low false negativity rate. Whilst Internet connectivity is currently essential for AI assessment, the rest of the functions such as patient enrolment or capturing images can be performed offline. This ensures that the images and demographic data of every participating woman is captured and stored digitally. The facility to provide immediate offline auto-assessment would be made available in the upcoming version of the software. A stumbling block in this study was AI missing out on two carcinoma cases. The reason behind this is the limited availability of validated images of carcinoma in the AI training set. In the future, we plan to strengthen AI algorithms by focusing on this category. Additionally, the validation of an individual category viz. amber or HRA will also be reported in future publications.

## 5. Conclusions

The AI-enabled Smart Scope^®^ test is an accurate digital VIA test that classifies cervical health in multiclass categories. It can be used as a point-of-care screening test as its sensitivity is 90.3%. This test can also be implemented as a point-of-care triaging test as its PPV is 83.58%. The accuracy of the AI-enabled Smart Scope^®^ test is 84.04% compared to that of colposcopy (59.15%). Since it is AI-enabled, the results are not subjective in nature. The Smart Scope^®^ test is user-friendly, can be performed by a minimally trained nurse, and based on the auto-assessment, it is possible to take an immediate decision regarding patient management in a minimalistic setting such as PHCs in LMICs.

## Figures and Tables

**Figure 1 diagnostics-13-03085-f001:**
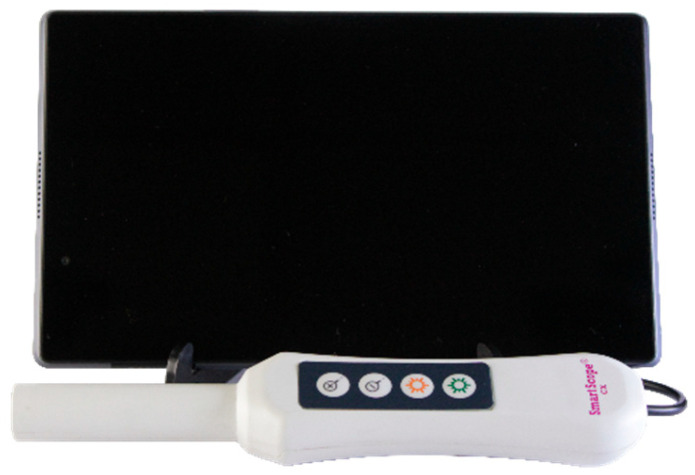
Transvaginal digital cervical health screening device Smart Scope^®^ along with a tablet.

**Figure 2 diagnostics-13-03085-f002:**
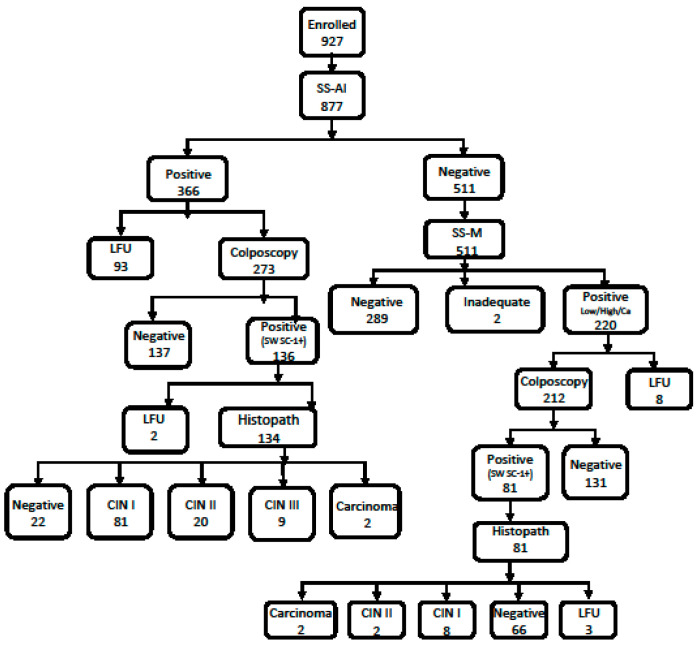
Histopathology outcome of the SS-AI screened women. LFU; loss to follow-up.

**Figure 3 diagnostics-13-03085-f003:**
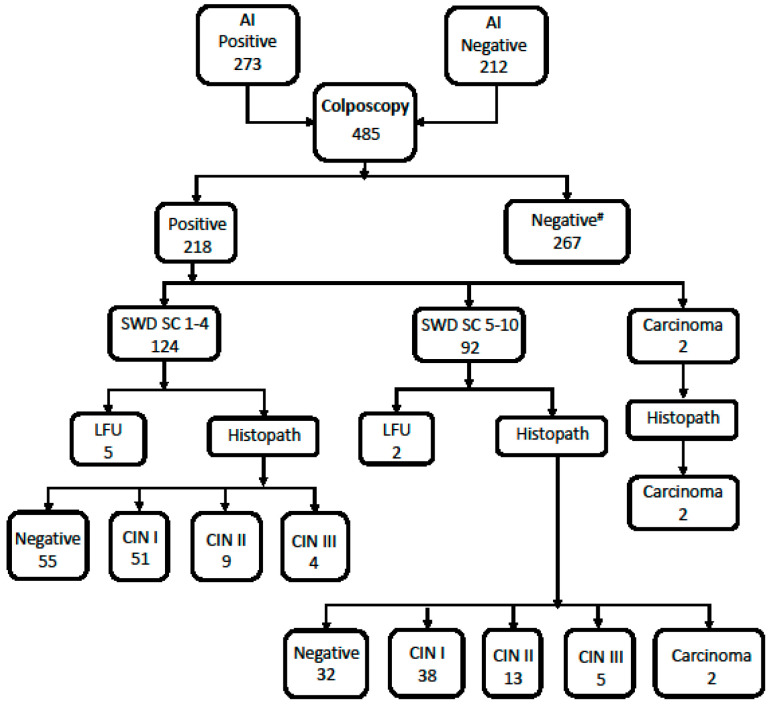
Histopathological outcome of women assessed by colposcopy. LFU; loss to follow-up.

**Figure 4 diagnostics-13-03085-f004:**
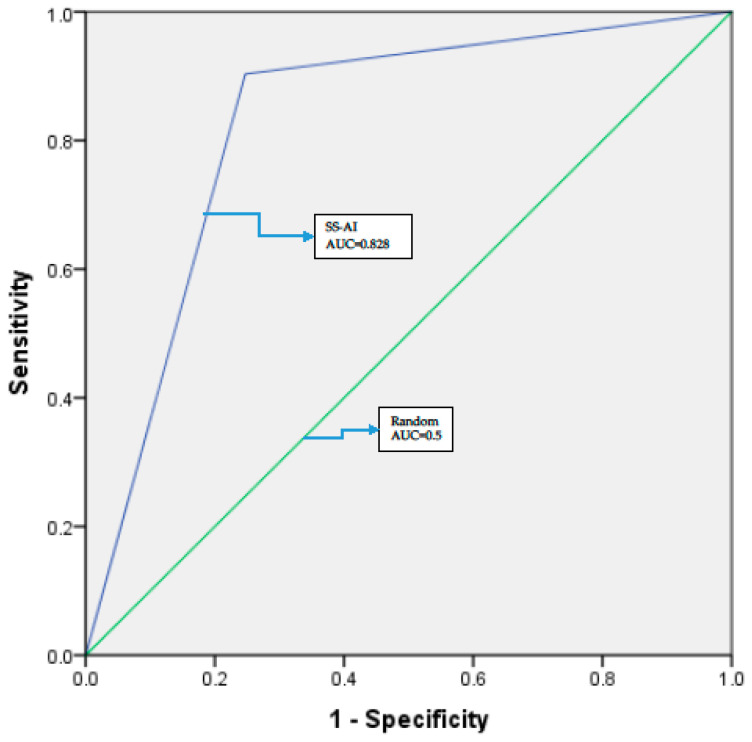
ROC curve. The blue line represents the AI enabled Smart Scope^®^ test with histopathology (CIN I+) as a reference test. The green line represents a random curve.

**Table 1 diagnostics-13-03085-t001:** Distribution of the demographic characteristics (*n* = 877).

Demographic Characteristics	No. of Women	Percentage	Demographics Characteristic	No. of Women	Percentage
Age (Years)			Family Income P.M.(Rs.)		
25–35	446	50.85	≤10,000	307	35
36–45	306	34.89	11,000–20,000	422	48.12
46–55	101	11.51	21,000–50,000	129	14.56
56–65	24	2.73	51,000≤	19	2.1
Education			Age at Marriage		
None	289	32.95	below 18 years	236	26.9
Primary	82	9.35	18 years	206	23.48
Middle	177	20.18	above 18 years	435	49.6
Highschool	179	20.41	Number of Deliveries		
Graduate	80	9.12	0	47	5.35
Postgraduate	70	7.98	1	106	12.08
Occupation			2	353	40.25
Housewife	859	97.94	3	240	27.37
Formal employment	18	2.05	4≤	131	14.94

**Table 2 diagnostics-13-03085-t002:** Distribution of the AI assessment based on the histopathology outcome (*n* = 213).

Histopathology	SS-AI			
	Green	Amber	HRA	Red
Normal (48)	14	24	5	5
Benign * (41)	2	27	1	11
CIN I (89)	2	6	30	51
CIN II (22)	0	2	10	10
CIN III (9)	0	0	7	2
Carcinoma (4)	0	2	2	0
**Total (213)**	**18**	**61**	**55**	**79**

*n*, Histopathology sample size; *, Includes cervicitis, squamous metaplasia, ectropion, polyp, inflammation, infection.

**Table 3 diagnostics-13-03085-t003:** Distribution of colposcopy assessment based on histopathology outcome (*n* = 213).

Histopathology	Colposcopy			
	Benign *	Swede Score 1–4	Swede Score 5–10	Carcinoma
Normal (48)	0	28	20	0
Benign * (41)	2	27	12	0
CIN I (89)	0	51	38	0
CIN II (22)	0	9	13	0
CIN III (9)	0	4	5	0
Carcinoma (4)	0	0	2	2
Total (213)	2	119	90	2

*n*, Histopathology sample size; *, Includes cervicitis, squamous metaplasia, ectropion, polyp, inflammation, infection.

## Data Availability

Data is unavailable due to privacy or ethical restrictions.

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
