# Peer review of "Automated Assessment of Digital Images of Uterine Cervix Captured Using Transvaginal Device—A Pilot Study"

_diagnostics, 2023, doi:10.3390/diagnostics13193085_

Round 1
Reviewer 1 Report
Dr. S. Shamsunder's manuscript investigates the potential diagnostic role of a digital device which, through artificial intelligence, can predict pre-neoplastic lesions of the uterine cervix. Although the reported data need subsequent validation, the results of the study are very encouraging. Moreover, the theme of artificial intelligence applied to medicine is of great importance for the scientific literature nowadays.
the overall quality of English is Good. Onlu minor typos are present within the text
Reviewer 2 Report
The article is original, well-written, with few English or typing errors (I have no expertise in the English language).
The article is objective, but some readers may find it too complex. Therefore, a small improvement in the fluidity of the text in the discussion could improve the understanding and scope of the content.
